# Validation Study of the PALCOM Scale of Complexity of Palliative Care Needs: A Cohort Study in Advanced Cancer Patients

**DOI:** 10.3390/cancers15164182

**Published:** 2023-08-20

**Authors:** Margarita Viladot, Jose-Luís Gallardo-Martínez, Fany Hernandez-Rodríguez, Jessica Izcara-Cobo, Josep Majó-LLopart, Marta Peguera-Carré, Giselle Russinyol-Fonte, Katia Saavedra-Cruz, Carmen Barrera, Manoli Chicote, Tanny-Daniela Barreto, Gemma Carrera, Jackeline Cimerman, Elena Font, Ignacio Grafia, Lucia Llavata, Javier Marco-Hernandez, Joan Padrosa, Anais Pascual, Dolors Quera, Carles Zamora-Martínez, Ana-Maria Bozzone, Carme Font, Albert Tuca

**Affiliations:** 1Unit of Supportive and Palliative Care in Cancer, Medical Oncology Department, Hospital Clínic de Barcelona, University of Barcelona, 08036 Barcelona, Spain; viladot@clinic.cat (M.V.); mbarrera@clinic.cat (C.B.); mchicote@clinic.cat (M.C.); barreto@clinic.cat (T.-D.B.); gecarrer@clinic.cat (G.C.); jacimerman@gmail.com (J.C.); efont@clinic.cat (E.F.); grafia@clinic.cat (I.G.); llavata@clinic.cat (L.L.); jmarco@clinic.cat (J.M.-H.); padrosa@clinic.cat (J.P.); alpascual@clinic.cat (A.P.); czamora@clinic.cat (C.Z.-M.); cfont@clinic.cat (C.F.); 2Home Care Support Teams Program (PADES) Group Mutuam, 08025 Barcelona, Spain; lui-jo@hotmail.com (J.-L.G.-M.); jessicaizcara@gmail.com (J.I.-C.); marta.peguera@gmail.com (M.P.-C.); ksaavedra36@gmail.com (K.S.-C.); 3Mutuam Güell Social Health Care Hospital, 08024 Barcelona, Spain; fhernandez@mutuam.com (F.H.-R.); gisele.russinyol@mutuam.com (G.R.-F.); dolors.quera@mutuam.com (D.Q.); 4ICO Gerona (Catalan Cancer Institute), 17007 Gerona, Spain; josep.majo@iconcologia.net; 5Psychosocial Support Team, “La Caixa” Foundation (EAPS), Hospital Clínic de Barcelona, 08036 Barcelona, Spain; 6catEAP Sarrià SLP, 08017 Barcelona, Spain; anaboz66.ab@gmail.com; 7Chair of Palliative Care, University of Barcelona, 08036 Barcelona, Spain

**Keywords:** early palliative care, advanced cancer, complexity of care needs, integration of palliative care in oncology, specialist palliative care, predictive model of complexity

## Abstract

**Simple Summary:**

There is sufficient evidence to confirm that early palliative care in advanced cancer patients improves symptom control, psychological distress, quality of life, patient and family satisfaction, futile use of cancer treatment at the end of life, use of healthcare resources and, in some cases, survival. In a patient-centred care model, referral to early palliative care depends on both prognosis and the complexity of care needs. Identifying the complexity of care needs is, therefore, key to appropriate referral to early palliative care teams. The PALCOM scale is a 5-domain multidimensional assessment tool developed to identify the level of complexity of the palliative needs of cancer patients. This validation study confirms the accuracy and predictive ability of the PALCOM scale to identify the level of complexity of palliative care needs in cancer patients. Higher levels of complexity are also associated with lower survival and higher in-hospital mortality.

**Abstract:**

Background: In a patient-centred model of care, referral to early palliative care (EPC) depends on both the prognosis and the complexity of care needs. The PALCOM scale is a 5-domain multidimensional assessment tool developed to identify the level of complexity of palliative care needs of cancer patients. The aim of this study was to validate the PALCOM scale. Patient and methods: We conducted a prospective cohort study of cancer patients to compare the PALCOM scale and expert empirical assessment (EA) of the complexity of palliative care needs. The EA had to categorise patients according to their complexity, considering that medium to high levels required priority attention from specialist EPC teams, while those with low levels could be managed by non-specialist teams. Systematically collected multidimensional variables were recorded in an electronic report form and stratified by level of complexity and rating system (PALCOM scale versus EA). The correlation rank (Kendall’s tau test) and accuracy test (F1-score) between the two rating systems were analysed. ROC curve analysis was used to determine the predictive power of the PALCOM scale. Results: A total of 283 advanced cancer patients were included. There were no significant differences in the frequency of the levels of complexity between the EA and the PALCOM scale (low 22.3–23.7%; medium 57.2–59.0%; high 20.5–17.3%). The prevalence of high symptom burden, severe pain, functional impairment, socio-familial risk, existential/spiritual problems, 6-month mortality and in-hospital death was significantly higher (*p* < 0.001) at the high complexity levels in both scoring systems. Comparative analysis showed a high correlation rank and accuracy between the two scoring systems (Kendall’s tau test 0.81, F1 score 0.84). The predictive ability of the PALCOM scale was confirmed by an area under the curve in the ROC analysis of 0.907 for high and 0.902 for low complexity. Conclusions: In a patient-centred care model, the identification of complexity is a key point to appropriate referral and management of shared care with EPC teams. The PALCOM scale is a high precision tool for determining the level of complexity of palliative care needs.

## 1. Introduction

The World Health Organisation (WHO) urges that health services be designed so that early palliative care is integrated into existing health services in a shared care model to provide appropriate support for all those who need it [1].

The early integration of palliative care into oncology is supported by many international scientific societies and expert commissions (American Society of Clinical Oncology [ASCO], European Society of Medical Oncology [ESMO], National Comprehensive Cancer Network [NCCN], Lancet Commission) [2,3,4,5,6,7]. Many randomised clinical trials have shown that early palliative care (EPC) in patients with advanced cancer improves symptom control, psychological distress, quality of life and patient and family satisfaction [8,9,10,11,12,13,14,15,16,17,18,19]. Some of these clinical trials have also shown that EPC reduce the use of futile chemotherapy at the end of life and improve healthcare resource use and survival [2,8,19].

Based on this consistent scientific evidence, the American Society of Clinical Oncology recommended that all patients with advanced cancer should be managed by multidisciplinary teams specialising in EPC in parallel with cancer treatment [20,21,22]. However, a consensus report from the Centre to Advance Palliative Care argued that it was neither desirable nor sustainable for all advanced patients to be cared for by specialised EPC teams, given the real limitations of these resources in daily clinical practice. Likewise, it warned of the urgent need to improve the basic training in palliative care of professionals attending severe patients with the aim of reserving specialised EPC services for problems surpassing their capacities [23].

Although most clinical trials support a systematic referral model to the EPC [8,9,11,17,19,24,25], a recent meta-analysis of lung cancer suggests that on-demand referral tailored to the needs of individual patients is as effective as systematic referral of all patients with advanced cancer [26]. Moreover, several studies also suggest that the EPC is more efficient in treating patients with a significant symptom burden and higher multidimensional needs [27,28].

In this context, the fundamental issue is no longer to accept the efficacy of EPC in advanced cancer patients but rather to determine the most efficient model for the referral and management of shared interventions in daily practice to ensure optimal palliative care for all patients in need. The disadvantage of the systematic referral of all advanced cancer patients, despite the proven effectiveness in most clinical trials, is that it requires a vast network of specialised EPC resources, which may not be easily accessible in real-world practice. Furthermore, such a system may provide unnecessary support to patients with low palliative care needs. The disadvantage of on-demand referral is that it depends on the subjective and often inconsistent empirical criteria of the referring professional, as well as the availability of specialised EPC resources. This can often result in late referrals and many patients missing the opportunity to receive personalised palliative care. It has now been suggested that targeted referral, a combination of both referral systems, may be an optimal model, as it would systematically use standardised criteria based on patient needs to prioritise and ensure early referral of all those patients most in need [29,30,31].

According to all these considerations, referral to specialised EPC teams within a shared intervention model depends not only on the state of evolution and prognosis of the patients but also on the complexity of their needs. The intrinsic difficulty of this argument, which is easy to accept from a theoretical point of view, is that the categorisation of the complexity of palliative needs is still not well defined.

Systemic theory defines a complex system as one that depends on the continuous adaptive interaction of multiple parts in a non-linear relationship and in an unstable equilibrium, highly sensitive to changes in initial conditions, where the outcome is difficult to predict and may be different and greater than the sum of the parts [32,33,34]. Translating this general definition into clinical practice, a patient would have more complex palliative care needs if his or her condition depended on multiple multidimensional variables interacting in an adaptive but unstable equilibrium, with an outcome that was not always predictable and required intensified supportive care [33,34,35,36,37,38].

A recent review of the literature identified 6 multidimensional evaluation tools aimed at defining and classifying the complexity of palliative needs [34]. These tools allow the identification of patients with low complexity, in whom basic palliative care by a non-specialised team would be indicated, and patients with medium-high complexity requiring the intervention of specialised EPC services. The PALCOM scale is one of these tools, which was specifically developed in 2018 for patients with advanced cancer [38].

The aim of this study was to validate the predictive capacity of the level of palliative complexity of the PALCOM scale. The hypothesis was that the PALCOM scale can reproduce with high accuracy the determination of the level of complexity of palliative needs assigned by experts.

## 2. Materials and Methods

A prospective, transversal, observational, multicentre study with a 6-month longitudinal follow-up of a cohort of patients with advanced cancer was conducted. 

### 2.1. Study Period and Setting

Several public healthcare centres of all levels of care (primary care, home care, hospital, medium-long stay units) of the Autonomous Community of Catalonia (Spain) participated in the study. 

The expert treatment team was multidisciplinary (doctors, nurses, social workers, psychologists). All had extensive experience in palliative care and were members of hospital supportive care in cancer units (inpatient and outpatient) or palliative care home teams. They also received specific training on how to optimise the registration of variables.

On the basis of the data recorded in the individual multidimensional assessment, the PALCOM scale was applied by a team of doctors who were external to the treating experts and not directly related to the patients enrolled.

The patient inclusion period was from December 2020 to April 2021 and the field work finalised 6 months after the inclusion of the last patient. 

### 2.2. Objectives and Main Variables 

The primary objective was to test the predictive and discriminative ability of the PALCOM scale in identifying levels of palliative care needs. The main variables were the value of the PALCOM scale and the level of palliative care complexity assigned by the treating professional using empirical assessment (EA). A comparison between the two variables allowed the identification of the predictive value of the PALCOM scale. The secondary objective was to describe the impact of the level of complexity on patient survival and hospital mortality. 

### 2.3. Inclusion Criteria

Patient inclusion was consecutive and included all patients fulfilling the following inclusion criteria: age ≥ 18 years, diagnosis of metastatic or advanced locoregional cancer, life expectancy ≤ 6 months, and signed informed consent by the patient.

### 2.4. Empirical Assessment of the Level of Complexity 

The EA was based on the following consensus definition: “The complexity of palliative care needs is a patient condition that depends on the individual interaction of emerging multidimensional characteristics that confer a particular tendency to instability, rapid change and uncertainty of the outcome of care and is associated with the need to intensify specialised care” [26].

Following multidimensional anamnesis, the treating professional determined the level of complexity of each patient (low, medium, high) according to their clinical judgement. A low level corresponded to needs that could be addressed by non-specialised teams, although it may sometimes be necessary to consult for the assessment or management of isolated symptoms. A medium-high level would require joint systematic care by specialised EPC teams with an intensity adapted to the level of complexity identified.

### 2.5. PALCOM Scale

The PALCOM scale is a tool for predicting the complexity of palliative care needs developed in 2018 specifically for patients with advanced cancer [26]. It is a tool consisting of 5 domains of multidimensional assessment: symptom burden, refractory pain, performance status, socio-familial risk, and existential/ethical issue (Figure 1). Each domain is scored dichotomously (0 = absent, 1 = present), with the total being the final value of the scale. This tool classifies the complexity of palliative needs into 3 levels: (a) Low, value 0–1, recommending basic palliative care, not necessarily shared with specialised EPC teams, although in some cases timely consultation with palliative care specialists may be required for comprehensive assessment or management of difficult isolated symptoms; (b) medium, value 2–3, recommending shared care with specialised EPC teams; (c) high, value 4–5, recommending intensive care by specialised EPC teams.

### 2.6. Registry of the Multidimensional Assessment

The following variables were systematically registered in an electronic case report form (e-CRF): sociodemographic data; primary origin/extension of cancer; oncological treatments in the last 4 weeks; Karnofsky index, intensity of 10 symptoms (pain, asthenia, anorexia, nausea-vomiting, constipation, dyspnoea, somnolence, insomnia, anxiety and depression), using a numerical rating scale (NRS) of 11 points; cognitive status, using the Confusion-Assessment-Method (CAM) observational tool [39,40], type of pain and identification of difficult pain according to the Edmonton Classification System for Cancer Pain (ECS-CP) [41], socio-familial risk factors and existential/ethical problems.

The Confusion Assessment Method (CAM) is an observational tool developed in 1990 to detect delirium. It is based on the criteria of the Diagnostic and Statistical Manual of Mental Disorders (DSM-5) [40] and has a sensitivity of over 90%. Several studies have confirmed its diagnostic accuracy in cancer patients [39].

The Edmonton Classification System for Cancer Pain (ECS-CP) is a tool consisting of five risk factors (pain mechanism, incidental pain, psychological status, history of addiction, cognitive status) that may predict analgesic response. The presence of neuropathic or mixed pain, incidental pain, psychological distress, addiction or cognitive impairment indicates that pain may be more difficult to control [41,42,43].

The variables were directly registered by the treating professional in a codified e-CRF, ensuring the confidentiality and protection of the personal data. 

### 2.7. Development of the Study 

The study was made up of the following successive stages: (1) Integral evaluation of the patient, systematic inclusion of the variables in the e-CRF and EA of the level of palliative complexity by the treating professional; (2) Determination of the level of complexity according to the PALCOM scale based on the e-CRF, by the central investigative team other than the treating professional; (3) Analysis of the accuracy and predictive power of the PALCOM scale; (4) Determination of survival and location of death in a follow-up of a maximum of 6 months after inclusion (Figure 2).

### 2.8. Statistical Analysis 

Taking into account the previous development study of the PALCOM scale (N = 324) [38] and the real enrolment capacity of the participating centres, we planned a study sample of at least 250 patients.

The categorical or dichotomic variables were analysed using absolute and relative frequencies. Continuous variables were described by calculating the central value (mean/median) and dispersion (standard deviation) with a 95% confidence interval (95% CI). The proportion of patients dying in the hospital was estimated using a binomial regression model. The function of survival was analysed using the Kaplan–Meier method. For the comparison between the different levels of complexity, we used the stratified test of logarithmic ranges and hazard ratios (HR) (95% CI) taken from the Cox model. Kendall’s Tau-b test was used as well as the F1 accuracy score to identify the grade of correlation of the level of palliative complexity assigned between the EA and the PALCOM scale. The reciever operating curves (ROC curve) was also used to assess the ability of the PALCOM scale to discriminate between high and low complexity.

## 3. Results

Seven healthcare centres participated in the study (3 home palliative care teams, 2 hospital services, 1 palliative care service in a medium-long stay unit, 1 primary care team). All the centres belonged to the public healthcare network of the urban areas of Barcelona and Gerona, Spain.

A total of 283 patients were included: 175 (61.8%) hospital; 65 (23.0%) home care teams; 22 (7.8%) medium-long stay units; 21 (7.4%) primary care. The mean age of the patients was 71.0 years (95% CI: 59.0–81.0) and 122 were males (56.9%). The most frequent primary origin of the cancer was the lung (24.4%). Metastatic extension was present in 77.7% of the patients and locoregional in 22.3%. In the four weeks prior to inclusion, 67.2% of the patients had received some specific cancer treatments (chemotherapy 47.7%, radiotherapy 12.7%, and hormone therapy 6.7%). The mean Barthel index was 66.7 (95% CI: 63.9–69.6) and the frequency of a Karnofsky index ≤ 50% was 47.5%.

At the baseline visit, the most frequent symptoms were asthenia (95.1%), pain (85.6%), anorexia (79.9%), sadness (69.3%), anxiety (65.0%), insomnia (62.5%), constipation (57.2%) and dyspnoea (39.2%). The intensity of pain was less moderate (NRS ≥ 4) in 66.9% of the patients. According to the ESC-CP, 58.5% of the patients with pain presented with characteristics of difficult pain. The intensity of anxiety and sadness was less moderate (NRS ≥ 4) in 43.0% and 51.6% of the patients, respectively. There was a high symptom burden (≥5 symptoms of at least moderate intensity, NRS ≥ 4) in 48.2% of the patients. The frequency of cognitive alterations detected using the CAM tool was 20.1%.

At least one factor of socio-familial risk was observed in 64.8% of the patients. The most frequent were limitation of support conditioned by advanced age or health problems of the main caregiver (35.3%); impossibility of management of the care needs and the occupational/economic burden of the main caregiver (20.4%); and lack of caregiver (18.4%). Some existential/ethical problems were observed in 23.6% of the patients at the baseline visit. The most frequent of these problems was dislike of the quality of information provided (14.6%), disagreement with the proportion of intensity of the healthcare intervention (4.6%) and the wish to advance death (3.9%) (Table 1).

A total of 278 patients completed the follow-up period (loss of contact in 5 cases). During the 6 months after inclusion, 185 patients (66.5%) died. The place of death was the home in 90 cases (48.6%), the hospital in 73 cases (39.4%), and the medium-long stay unit in 22 cases (11.9%).

### 3.1. Results According to the Level of Complexity 

There were no significant differences in the frequencies of the levels of palliative complexity assigned by the EA and the PALCOM scale (low 22.3 and 23.7%; medium 57.2 and 59.0%; high 20.5 and 17.3%, respectively). 

The prevalence of the symptoms systematically registered was significantly higher in the patients with greater complexity in both evaluation systems (Table 2). 

The prevalence of the 5 domains of the PALCOM scale (high symptom burden, difficult pain according to the ESC-CP, Karnofsky index ≤ 50%, socio-familial risk and existential/ethical problems) was significantly higher (*p* < 0.001) in the patients classified as high complexity compared with those of medium and low complexity in both evaluation systems (Table 3). The prevalence of high symptom burden in high complexity cases according to EA and the PALCOM scale was 77.6 and 95.9%, respectively, significantly higher (*p* < 0.001) than that observed in low (9.5–9.0%) and medium (52.5–49.7%) complexity cases. The prevalence of refractory pain in high complexity cases according to EA and the PALCOM scale was 79.3% and 87.7%, respectively, significantly higher (*p* < 0.001) than that observed in low (46.0–44.8%) and medium (55.8–55.4%) complexity cases. The prevalence of functional impairment, defined as KPS ≤ 50%, in high complexity cases according to EA and the PALCOM scale was 82.8% and 83.7%, respectively, significantly higher (*p* < 0.001) than that observed in low (9.5–6.0%) and medium complexity (49.7–53.6%). The prevalence of any sociofamily risk factor in high complexity cases according to EA and the PALCOM scale was 86.2% and 91.8%, respectively, which was significantly higher (*p* < 0.001) than that observed in low (27.0–19.4%) and medium complexity (71.8–75.0%) cases. The prevalence of high symptom burden in high complexity cases according to EA and the PALCOM scale was 77.6 and 95.9%, respectively, significantly higher (*p* < 0.001) than that observed in low (9.5–9.0%) and medium complexity (52.5–49.7%) cases. The prevalence of existential/ethical dilemmas in high complexity cases according to EA and the PALCOM scale was 44.8 and 59.2%, respectively, significantly higher (*p* < 0.001) than that observed in low (12.7–6.0%) and medium complexity (20.2–20.2%) cases.

The probability of death during the 6 months of follow-up was significantly higher (*p* < 0.001) in patients classified as high complexity (82.8% and 79.6% according to the EA and PALCOM scale, respectively) compared to those with medium (65.4% and 67.7% according to the EA and PALCOM scale, respectively) and low complexity (49.2% and 49.3% according to the EA and PALCOM scale, respectively) (Table 3). Kaplan–Meier actuarial survival at 6 months was significantly lower in high complexity cases compared with medium or low complexity cases in both assessment cohorts (EA and PALCOM scale) (*p* < 0.001) (Figure 3).

Death in hospital was significantly more frequent (*p* > 0.05) in patients with high complexity (25.0% and 30.8% according to the EA and PALCOM scale, respectively) and medium complexity (24.5% and 22.1% according to the EA and PALCOM scale, respectively) than in patients with low complexity (16.1% and 18.2% according to the EA and PALCOM scale, respectively).

### 3.2. Correlation between the Empirical Assessment and the PALCOM Scale 

Kendall’s tau test is a coefficient that measures the correlation between two columns of ranked data. The value of a correlation coefficient can range from −1 to 1, where −1 indicates a perfect negative relationship, 0 indicates no relationship and 1 indicates a perfect positive relationship. In this study, Kendall’s tau-b correlation coefficient between the EA and the PALCOM scale was 0.81 in this study.

The sensitivity (probability that a positive result is actually positive) of the PALCOM scale for low, medium and high palliative complexity was 0.89, 0.96 and 0.74, respectively, while the specificity (probability that a negative result is actually negative) was 0.69, 0.55 and 0.63, respectively. The negative predictive value of the PALCOM scale for low, medium and high palliative complexity was 0.78, 0.71 and 0.67, respectively. 

The F1 score measures the accuracy of a model on a dataset in a binary classification system. Determines the number of predictions that are correct relative to the total number of predictions made (the closer to 1, the higher the accuracy). The accuracy of the PALCOM model in relation to EA was confirmed by the identification of an F1 score of 0.84 in this study.

The receiver operating curve (ROC curve) is a graphical representation of the diagnostic ability of a binary classifier system by plotting the true positive rate against the false positive rate. The accuracy of the measurement increases the closer the ROC curve is to the upper left corner of the graph. The area under the ROC curve (AUC) is the area of the graph that lies below the distribution of scores, so a value of 0.5 would represent zero accuracy and a value close to 1 would be an optimal score. In this study, the ROC analysis showed a high ability of the PALCOM scale to discriminate between high and medium-low complexity of palliative care needs (AUC 0.907) and between low and high-medium complexity (AUC 0.902) (Figure 4).

## 4. Discussion

We consider that the sample was consistent because both the sociodemographic and clinical data, as well as the distribution of frequencies of complexity, were comparable to those of previous studies [38]. The correlation between the higher level of complexity and the increase of the prevalence of the multidimensional variables, 6-month mortality and hospital death confirm the coherence of the assessment model. Likewise, a greater prevalence of the 5 domains of the PALCOM scale was associated with high levels of complexity. The data concordance between observers confirms the consistency of the EA in determining the level of complexity assigned by the experts after a multidimensional assessment. The high correlation between the EA and the values provided by the PALCOM scale confirms that this instrument has a high capacity to discriminate the level of complexity of palliative needs. Its use, whether by specialised professionals or not, can provide a homogeneous assessment and categorisation model of the level of complexity that is fully comparable with the opinions of experts.

The data from the PALCOM development [38] and validation studies met the criteria of the complexity construct from systemic theory [30,31,32,33,34,35,36,37,44]. In fact, in the PALCOM scale, the level of complexity is associated with the dynamic interaction of multidimensional variables in an unstable balance, rather than with the intensity of a symptom or condition assessed in isolation.

Most experts and clinical guidelines recommend an integrated model of care for patients with advanced cancer that includes early referral to specialised palliative care teams [20,21,22,45]. Many of these experts and guidelines recommend systematic referral to the EPC of all patients with incurable advanced cancer and limited survival soon after diagnosis and concurrently with cancer treatment. However, there are two issues raised by some experts that remain unresolved. The first is whether it is better to refer all patients than to prioritise those with the greatest need. The second is whether systematic referral of all patients is more effective than referral on an as-needed basis. In 2011, the Centre for Advance Palliative Care argued that it was neither desirable nor sustainable for all advanced patients to be cared for by specialist EPC teams and suggested that those with higher needs at any point in the course of their incurable disease should be prioritised [23]. Three systematic reviews of the literature confirm that the effectiveness of the EPC intervention is similar for systematic and on-demand referral at any point in the course of the disease, based on the specific needs of the patient [25,26,46]. Given these arguments and the imperative to provide high-quality palliative care to all patients who need it in a context where healthcare resources are not unlimited, it would be reasonable to consider a model of referral to EPC based on the complexity of care needs. A shared care model based on complexity should include the training of oncology professionals (physicians, advanced practice nurses) in basic multidimensional assessment and complexity screening, so that specialised EPC teams can focus on patients with more complex needs. A key issue and the main difficulty of this model is how to define and categorise the complexity of the patient’s needs.

Different models of classification of the complexity of palliative needs have been developed (Hex-Com, Perroca-Scale, AN-SNAP, Hui-Major-Criteria, IDC-Pal, PALCOM) [35,35,37,47,48,49,50,51,52,53,54,55,56,57,58,59,60]. Most of these tools are based on the consensus of experts who identified the characteristics considered to be of greatest complexity and their posterior validation in cross-sectional or Delphi studies. The design of the PALCOM study was different. This was a prospective study of a cohort of patients with advanced cancer using a systematic multidimensional assessment based on the theoretical model of complexity and an empirical assignment of complexity by an expert, without prior consensus on the variables of greater or lesser complexity. The logistic regression analysis identified the highly discriminatory variables of the level of complexity and allowed the construction of the predictive model of the PALCOM scale [38]. Therefore, the aim of the PALCOM scale was not to describe the construct of complexity but to develop a categorisation model of the complexity of palliative care needs using only those variables that proved to be highly discriminative in the model. The current study was conducted using the same methodology and provided external validation of the high discriminative power of this scale.

### Study Limitations

This study only included patients with advanced cancer within a large public healthcare network; therefore, we cannot confirm the usefulness of this scale in people with chronic non-cancer diseases and/or in a different care setting. 

As the PALCOM scale was developed for adult patients, it is not applicable to paediatric palliative care without a specific validation study. 

Patients with an expected survival of 6 months or less were included in order to shorten the follow-up period of the study. There is no reason to believe that the PALCOM scale is not applicable to patients with longer survival.

By definition, complexity is a dynamic process. However, the data presented in this study expose the determination of the level of complexity made at the patient’s admission visit and do not report on the patient’s evolution during follow-up. We are now conducting a secondary analysis of aggregated data from the PALCOM development and validation studies to determine the dynamic behaviour of the complexity along the evolution of the patient and its impact on healthcare resource use.

## 5. Conclusions

The data from this study validate the PALCOM scale as a tool with a high capacity to discriminate the level of the complexity of palliative needs, which can be used by both specialised and non-specialised professionals. 

Practical implications:

In a patient-centred shared care model, it is essential to have a tool to optimise early referral and management of the intensity of EPC team intervention. The aim is to ensure that no patient loses the opportunity to receive high-quality palliative care, that care is prioritised for those patients who need it most, and that unnecessary specialist resources are not activated. The PALCOM scale is a practical and effective tool for healthcare professionals to assess palliative care needs in a homogeneous and consistent way.

In this context, we believe that integrating the assessment of complexity of needs into care pathways, using tools such as the PALCOM scale, can make a significant contribution to improving care and consensual management of shared care. There is a need to strengthen the training of non-specialist professionals, both in basic palliative care for low-complexity patients and in tools or criteria of complexity, to ensure that referral to the EPC is tailored to the needs of the patient.

Research implications:

Further research should explore the applicability of the PALCOM scale in different healthcare settings and in non-cancer palliative care. Longitudinal studies can explore the dynamic nature of complexity over time, which may help optimise care. Comparative studies of different referral approaches using the scale would provide valuable insights.

In conclusion, the PALCOM scale may represent a significant advance in palliative care practice. Its implementation has the potential to improve patient outcomes, optimise resource allocation and enhance patient-centred palliative care.

## Figures and Tables

**Figure 1 cancers-15-04182-f001:**
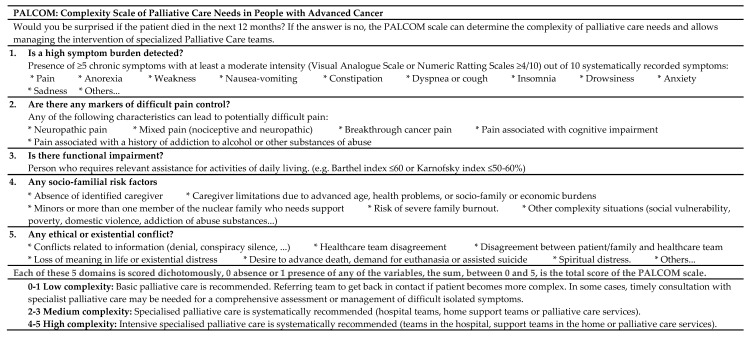
PALCOM Scale.

**Figure 2 cancers-15-04182-f002:**
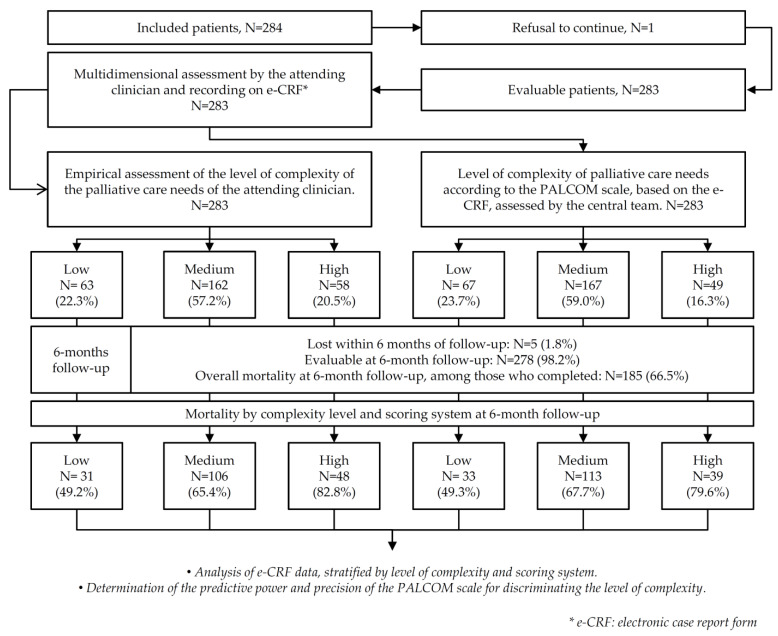
PALCOM validation study flow chart.

**Figure 3 cancers-15-04182-f003:**
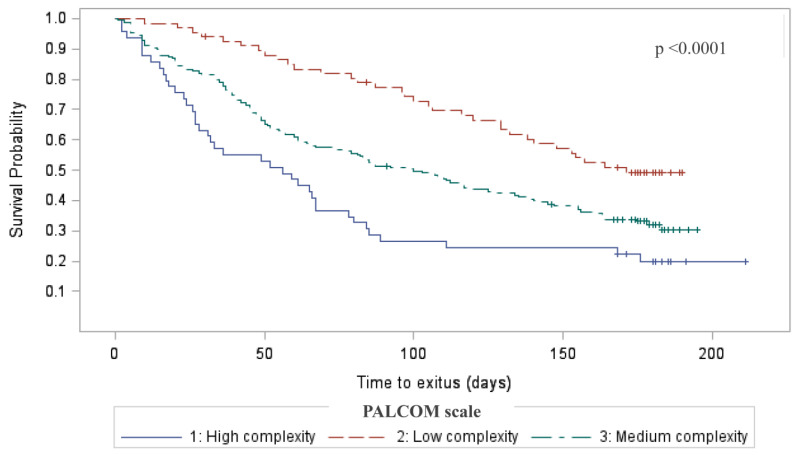
Actuarial survival by PALCOM level of complexity of palliative care needs (Kaplan-Meier).

**Figure 4 cancers-15-04182-f004:**
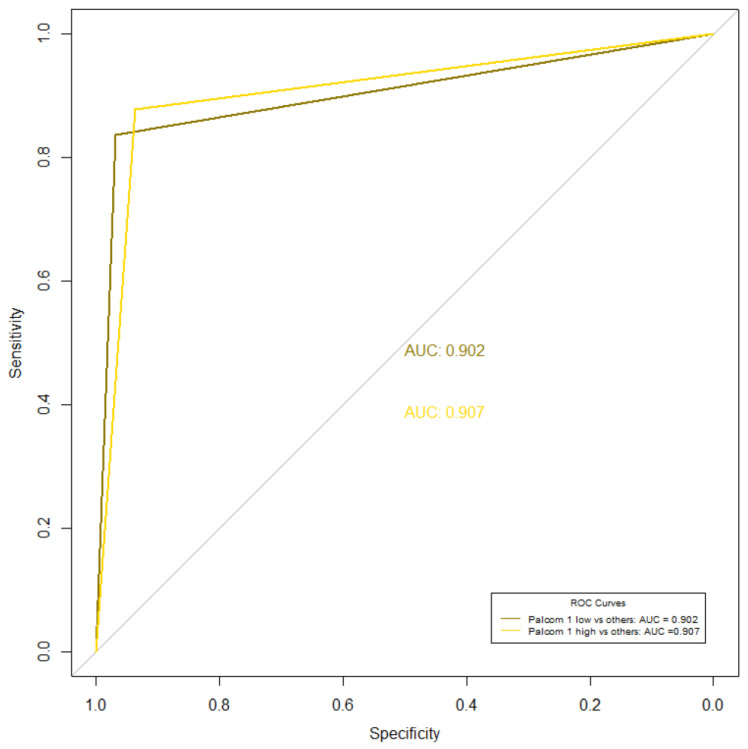
The receiver operating curve (ROC curve) of PALCOM scale.

**Table 1 cancers-15-04182-t001:** Overall characteristics of patients included in the PALCOM study.

			N	%
N total			283	100
Gender	Men		161	56.9
Age	Mean (SD)	71 (SD ± 59.0 − 81.0)		
Primary cancer	Lung		69	24.4
	Colon		56	19.8
	Prostate		28	9.9
	Breast		19	6.7
	Pancreas		18	6.4
	Other origins (<5%) *		93	32.8
Cancer extension	Local and regional		63	22.3
	Metastases		220	77.7
Cancer treatment, last 4 weeks	Overall		190	67.2
		Chemotherapy **	135	47.7
		Radiotherapy	26	12.7
		Hormonal therapy	19	6.7
Karnofsky index	≤50%		135	47.5
Symptoms prevalence	Asthenia		269	95.1
	Anorexia		226	79.9
	Pain		245	86.6
	Nausea		68	24.0
	Constipation		162	57.2
	Dyspnoea		111	39.2
	Insomnia		177	62.5
	Anxiety		184	65.0
	Sadness		196	69.3
	No well-being sense		273	96.5
High symptom burden	≥5 symptoms with intensity in NRS ≥ 4.		136	48.1
Pain characteristics	Type	Nociceptive somatic	149	60.8
		Nociceptive visceral	107	43.7
		Neuropathic	73	29.8
	Difficult pain according ECS-CP ***		166	58.5
		Breakthrough pain	135	55.1
		Psychological distress	64	26.1
		Addictive behaviour	20	8.2
		Cognitive impairment	26	9.3
		Mixed pain ****	73	29.8
Social risk factors according PALCOM scale			184	64.8
Existential/ethical problems according PALCOM scale			67	23.6

SD: Standard deviation. * Other primary origins with a frequency lower than 5%. ** Chemotherapy or other systemic treatments different from hormonal treatment (immunotherapy, targeted therapies…). *** ECS-CP: Edmonton Classification System Cancer Pain [19]. **** Any pain with a neuropathic component.

**Table 2 cancers-15-04182-t002:** Characteristics of patients according to level of palliative complexity and assessment method (professional empirical assessment and PALCOM scale).

		Professional Empirical Assessment N (%)				PALCOM Scale N (%)			
		Low	Medium	High		Low	Medium	High	
N total = 283		63 (22.3)	162 (57.2)	58 (20.5)		67 (23.7)	167 (59.0)	49 (17.3)	
Gender: Men		40 (63.5)	92 (56.8)	29 (50.0)		41 (61.2)	97 (58.1)	23 (46.9)	
Cancer treatment last 4 weeks		36 (57.1)	91 (56.2)	23 (39.7)		41 (61.2)	87 (52.1)	22 (44.9)	
Symptom prevalence					*p*				*p*
	Asthenia	57 (90.5)	154 (95.1)	58 (100)	<0.0001	62 (92.5)	158 (94.6)	49 (100)	<0.0001
	Anorexia	40 (63.5)	134 (82.7)	52 (89.7)	<0.0001	45 (67.2)	137 (82.0)	44 (89.8)	<0.0001
	Pain	50 (79.4)	140 (86.4)	53 (91.4)	<0.0001	54 (80.6)	140 (83.8)	49 (100)	<0.0001
	Nausea	10 (15.9)	37 (22.8)	21 (36.2)	0.032	11 (16.4)	35 (21.0)	22 (44.9)	0.007
	Constipation	25 (39.7)	95 (58.4)	42 (72.4)	0.003	28 (41.8)	98 (58.7)	36 (73.5)	<0.0001
	Dyspnoea	16 (25.4)	61 (37.7)	34 (58.6)	0.017	18 (26.9)	62 (37.1)	31 (63.3)	0.004
	Insomnia	27 (42.9)	106 (65.4)	44 (75.9)	<0.0001	30 (44.8)	107 (64.1)	40 (81.6)	<0.0001
	Anxiety	24 (38.1)	114 (70.4)	46 (79.3)	<0.0001	30 (44.8)	111 (66.5)	43 (87.8)	<0.0001
	Sadness	34 (54.0)	116 (71.6)	46 (79.3)	<0.0001	35 (52.2)	117 (70.1)	44 (89.8)	<0.0001
	No well-being sense	61 (96.7)	157 (96.1)	55 (94.8)	<0.0001	66 (98.5)	161 (96.4)	46 (93.9)	<0.0001
Karnofsky index ≤ 50%		6 (9.5)	81 (49.7)	48 (82.8)	<0.0001	4 (6.0)	90 (53.6)	41 (83.7)	<0.0001
Factors of social-family risk		17 (27.0)	117 (71.9)	50 (86.2)	<0.0001	13 (19.4)	126 (75.0)	45 (91.8)	<0.0001
Existential/ethical problems		8 (12.7)	33 (20.2)	26 (44.8)	<0.0001	4 (6.0)	34 (20.2)	29 (59.2)	<0.0001

**Table 3 cancers-15-04182-t003:** Complexity levels of PC needs, according to empirical assessment of attending professional and PALCOM scale.

	Professional Empirical Assessment N (%)			PALCOM Scale N (%)			
	Low	Medium	High	Low	Medium	High	*p*
N total = 283	63 (22.3)	162 (57.2)	58 (20.5)	67 (23.7)	167 (59.0)	49 (17.3)	NSD
PALCOM domains							
▪ High symptom burden ▪ Refractory pain ▪ KPS ≤ 50% ▪ Social risk ▪ Existential/ethical dilemma	6 (9.5)	85 (52.5)	45 (77.6)	6 (9.0)	83 (49.7)	47 (95.9)	<0.001
	29 (46.0)	91 (55.8)	46 (79.3)	30 (44.8)	93 (55.4)	43 (87.8)	<0.001
	6 (9.5)	81 (49.7)	48 (82.8)	4 (6.0)	90 (53.6)	41 (83.7)	<0.001
	17 (27.0)	117 (71.8)	50 (86.2)	13 (19.4)	126 (75.0)	45 (91.8)	<0.001
	8 (12.7)	33 (20.2)	26 (44.8)	4 (6.0)	34 (20.2)	29 (59.2)	<0.001
Death before 6 months of follow-up. N = 185 (65.4)	31 (49.2)	106 (65.4)	48 (82.8)	33 (49.3)	113 (67.7)	39 (79.6)	<0.001
Lost within 6 months of follow-up. N = 5 (1.8)	0	5 (3.0)	0	2 (2.9)	3 (1.8)	0	-
Hospital death	5 (16.1)	26 (24.5)	12 (25)	6 (18.2)	25 (22.1)	12 (30.8)	<0.05

NSD: no statistical differences in overall determination of complexity level between empiric assessment and PALCOM scale. KPS: Karnofsky index.

## Data Availability

The data can be shared upon request.

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
