# Peer review of "Validation Study of the PALCOM Scale of Complexity of Palliative Care Needs: A Cohort Study in Advanced Cancer Patients"

_cancers, 2023, doi:10.3390/cancers15164182_

Round 1

Reviewer 1 Report

Well done; a well written paper addressing a challenging issue in palliative care

There is a bit of overlap in the introduction and discussion around blanket vs targetted referral to PC; probably reducing the intro would be helpful

Author Response

Dear Reviewer,

Thank you very much for your comments.

Following your suggestions, we have modified the introduction to avoid overlap with the discussion. However, please note that we have added some concepts suggested by the other reviewer to the introduction.

Once again, we are very grateful for your contribution to our manuscript.

Yours sincerely.

Albert Tuca

Reviewer 2 Report

The manuscript titled "Validation study of the PALCOM scale of complexity of palliative care needs: a cohort study in advanced cancer patients" addresses an important topic related to identifying and validating an assessment tool for palliative care needs in cancer patients. The study's aim and objectives are clearly stated in the abstract, providing a concise overview of the research.

The introduction provides a comprehensive background on the importance of early palliative care (EPC) integration into oncology and the need for an assessment tool to identify the complexity of palliative care needs. The references cited support the arguments made, and the introduction sets the stage for the study's objectives. However, some aspects could be improved to enhance its clarity and effectiveness.

1.          Clarity of Purpose: The introduction lacks a clear statement of the study's objective. While it mentions the validation of the predictive capacity of the PALCOM scale, a more explicit research question or hypothesis would improve the introduction's focus.

2.          Additional References: The introduction could benefit from additional references to support the claims made. Including more recent literature or critical studies on early palliative care and the significance of assessing palliative care complexity would strengthen the foundation of the research.

3.          Definition of Terms: The introduction introduces the term "palliative complexity," which may not be immediately clear to all readers. A brief definition or explanation of this term and its significance would improve the introduction's accessibility.

Overall, the introduction provides relevant information, but it can be improved by clearly stating the study's specific objective, including additional supporting references, defining key terms, and providing more contextual background.

Methods

The materials and methods section describes the study design, patient cohort, data collection tools, and statistical analysis. Overall, the research design appears appropriate for the objectives of the study. However, there are areas where further improvements can be made to enhance clarity and completeness.

·       Sample Size Justification: The authors mention that the sample size of at least 250 patients was estimated based on the study that generated the PALCOM scale and the inclusion capacity of the participating centers. Additional details on this sample size's rationale or statistical considerations would help understand its appropriateness.

·       Data Collection: The data collection tools and variables description is sufficiently detailed. However, including a reference or providing more information on the Confusion-Assessment-Method (CAM) observational tool and the Edmonton Classification System for Cancer Pain (ECS-CP) would benefit unfamiliar readers.

·       Training and Consistency: Since multiple healthcare centers participated in the study, it would be valuable to include information about how the professionals were trained to ensure consistent and standardised data collection across centers.

Results

The results section presents the findings of the study clearly and concisely. The authors provide detailed information on the study cohort, the prevalence of symptoms and complexity levels, correlations between empirical assessment and the PALCOM scale, and the predictive ability of the PALCOM scale. However, some areas can be improved to enhance the clarity and interpretation of the results.

·       Data Presentation: The results are primarily presented in tables and figures, which help visualise the data. However, there could be more direct references to the specific tables or figures in the text, allowing readers to associate the text with the corresponding data quickly.

·       Statistical Interpretation: While the results include statistical measures such as p-values, sensitivity, specificity, and AUC, it would be beneficial to include a brief explanation of these measures and their implications to help readers understand the statistical significance and the predictive power of the PALCOM scale.

Discussion and conclusion

The discussion is well-supported by the study results, and the authors comprehensively analyse the findings. They highlight the consistency of the sample, the correlation between complexity levels and multidimensional variables, mortality, and hospital deaths. The discussion also includes comparisons with previous literature, addressing unresolved issues regarding systematic versus on-demand referral to specialised palliative care teams.

The authors elaborate on the development and validation of the PALCOM scale, emphasising its high discriminative power and ability to categorise the complexity of palliative care needs effectively. They discuss the benefits of using the scale in a patient-centred care model, allowing for early referral to specialised palliative care teams and better management of shared interventions.

The conclusion is a concise summary of the study's findings and the validation of the PALCOM scale as a valuable tool for discriminating the complexity of palliative needs. It emphasises the scale's applicability to both specialised and non-specialised professionals. It underscores the importance of using validated tools for assessing the complexity of care needs in a patient-centred approach.

Overall, the discussion and conclusions are well-supported and provide valuable insights into the implications of the study's findings.

Overall, your study on the PALCOM scale, a tool for assessing the complexity of palliative care needs in patients with advanced cancer, is commendable and provides valuable insights for palliative care practice and research. The results demonstrate that the PALCOM scale has a high capacity to discriminate the level of complexity, and its user-friendly nature allows both specialised and non-specialised professionals to use it effectively.

Some suggestions to improve your conclusion:

·       Practical Implications: The validated PALCOM scale offers a practical and effective tool for healthcare professionals to assess palliative care needs. Early referral to specialised palliative care teams based on the scale's assessment can improve patient outcomes and resource optimisation. Implementing a shared care model based on complexity can ensure appropriate care allocation.

·       Research Implications: Further research should explore the applicability of the PALCOM scale in diverse healthcare settings and non-cancer palliative care. Longitudinal studies can examine the dynamic nature of complexity over time, aiding care optimisation. Comparative studies on referral approaches using the scale would provide valuable insights.

·       Clinical Implementation: I recommend integrating the PALCOM scale into palliative care pathways and providing training programs for healthcare professionals. Regular audits and quality improvement initiatives will ensure its successful Implementation and continuous enhancement of patient care.

·       Patient-Centred Care: Adopting the PALCOM scale aligns with patient-centred care principles. Involving patients and families in the complexity assessment process enhances shared decision-making and patient satisfaction.

In conclusion, the PALCOM scale significantly advances palliative care practice. Its Implementation can potentially improve patient outcomes, optimise resource allocation, and enhance patient-centred palliative care.

Author Response

Dear reviewer,

Thank you very much for your comments.

Following your suggestions we have modified the manuscript as follow:

  1. Clarity of Purpose: The introduction lacks a clear statement of the study's objective. While it mentions the validation of the predictive capacity of the PALCOM scale, a more explicit research question or hypothesis would improve the introduction's focus.

Answer: More explicit information about the aims, research question and hypothesis has been added to the introduction.

  1. Additional References: The introduction could benefit from additional references to support the claims made. Including more recent literature or critical studies on early palliative care and the significance of assessing palliative care complexity would strengthen the foundation of the research.
  2. Answer: More updated references have been added to improve the research arguments. Particular emphasis has been placed on referral methods and the new concept of targeted referral.
  3. Definition of Terms: The introduction introduces the term "palliative complexity," which may not be immediately clear to all readers. A brief definition or explanation of this term and its significance would improve the introduction's accessibility.

Answer: We have included a general review of the construct and definition of complexity since systems theory, as well as references to respect.

  1. Sample Size Justification: The authors mention that the sample size of at least 250 patients was estimated based on the study that generated the PALCOM scale and the inclusion capacity of the participating centers. Additional details on this sample size's rationale or statistical considerations would help understand its appropriateness.

Answer: The statisticians who advised us reported that there is no clearly identified method for sample size in an observational prevalence study. For this reason, they suggested that a sample size similar to that used in the developmental study should be used. They argued that when designing a study of these characteristics, it is more important to avoid selection bias and to ensure registration and monitoring of variables than to speculate about the size of the study. They suggested the maximum number possible, taking into account the real setting of the study, and always a rigorous protocol and a size very close to the previous study.

  1. Data Collection: The data collection tools and variables description is sufficiently detailed. However, including a reference or providing more information on the Confusion-Assessment-Method (CAM) observational tool and the Edmonton Classification System for Cancer Pain (ECS-CP) would benefit unfamiliar readers.

Answer: We have included more information and references about Confusion-Assessment-Method and the Edmonton Classification System for Cancer Pain.

  1. Training and Consistency: Since multiple healthcare centers participated in the study, it would be valuable to include information about how the professionals were trained to ensure consistent and standardized data collection across centers.

Answer: We have expanded the information on the characteristics and specific training of the professionals participating in the study.

  1. Data Presentation: The results are primarily presented in tables and figures, which help visualise the data. However, there could be more direct references to the specific tables or figures in the text, allowing readers to associate the text with the corresponding data quickly.

Answer: We have expanded the data on the results, presented in tables, in the body of the manuscript.

  1. Statistical Interpretation: While the results include statistical measures such as p-values, sensitivity, specificity, and AUC, it would be beneficial to include a brief explanation of these measures and their implications to help readers understand the statistical significance and the predictive power of the PALCOM scale.

Answer: We have added information about statistical methods and their interpretation in this study.

  1. Some suggestions to improve your conclusion.

Answer: We have modified the conclusions according to your suggestions.

Once again, thank you very much for your kind comments, which have certainly improved our work.

Yours sincerely

Albert Tuca
